# Magnetic Response of Nano/Microparticles into Elastomeric Electrospun Fibers

**DOI:** 10.3390/jfb14020078

**Published:** 2023-01-29

**Authors:** Vincenzo Iannotti, Giovanni Ausanio, Anna M. Ferretti, Zaheer Ud Din Babar, Vincenzo Guarino, Luigi Ambrosio, Luciano Lanotte

**Affiliations:** 1CNR-SPIN and Department of Physics “E. Pancini”, University of Naples Federico II, Piazzale V. Tecchio 80, 80125 Naples, Italy; 2Istituto di Scienze e Tecnologie Chimiche “Giulio Natta” (SCITEC), Consiglio Nazionale delle Ricerche, Via G. Fantoli 16/15, 20138 Milan, Italy; 3Scuola Superiore Meridionale (SSM), University of Naples Federico II, Largo S. Marcellino, 10, 80138 Naples, Italy; 4Institute of Polymers, Composites and Biomaterials (IPCB), National Research Council of Italy, Mostra d’Oltremare Pad. 20, V.le J.F. Kennedy 54, 80125 Naples, Italy

**Keywords:** iron oxide nanoparticles, nickel particles, magnetic-functionalized fibers, electrospinning, magnetic nanocomposite

## Abstract

Combining magnetic nanoparticles (MNPs) with high-voltage processes to produce ultra-thin magnetic nanofibers (MNFs) fosters the development of next-generation technologies. In this study, polycarbonate urethane nanofibers incorporating magnetic particles were produced via the electrospinning technique. Two distinct types of magnetic payload were used: (a) iron oxide nanoparticles (IONPs) with an average size and polydispersity index of 7.2 nm and 3.3%, respectively; (b) nickel particles (NiPs) exhibiting a bimodal size distribution with average sizes of 129 nanometers and 600 nanometers, respectively, and corresponding polydispersity indexes of 27.8% and 3.9%. Due to varying particle sizes, significant differences were observed in their aggregation and distribution within the nanofibers. Further, the magnetic response of the IONP and/or NiP-loaded fiber mats was consistent with their morphology and polydispersity index. In the case of IONPs, the remanence ratio (M_r_/M_s_) and the coercive field (H_c_) were found to be zero, which agrees with their superparamagnetic behavior when the average size is smaller than 20–30 nm. However, the NiPs show M_r_/M_s_ = 22% with a coercive field of 0.2kOe as expected for particles in a single or pseudo-single domain state interacting with each other via dipolar interaction. We conclude that magnetic properties can be modulated by controlling the average size and polydispersity index of the magnetic particles embedded in fiber mats to design magneto-active systems suitable for different applications (i.e., wound healing and drug delivery).

## 1. Introduction

Among different nanomaterials, magnetic nanoparticles (MNPs) have a variety of applications in various fields, including hyperthermia therapy [1,2], contrast agents in magnetic resonance imaging [3,4], and biosensing [5,6]. Recent studies demonstrated that electrospun fibers with embedded nanoparticles exhibit new functionalities at the nanoscale due to the optimization of process conditions [7,8], thus, showing improved performance and increased protection of nanoparticles from oxidation [9]. Similarly, magnetic nanoparticles embedded in nanofibers exhibit intriguing features [10] and offer magnetic-field-dependent mechanical properties [11]. Such magnetically responsive materials can be used as “smart” fibers in healthcare, such as bio-inspired membranes for wound healing [12], tissue engineering [13], sensors and actuators [14,15], magnetic hyperthermia in cancer treatment [16], and controlled drug release [17]. The physical properties of functionalized magnetic nanofibers (MNFs) can be tuned by incorporating different nanoparticles with specific magnetic properties and responses [7].

Moreover, the properties of such functionalized MNFs depend on (i) morphological structure, (ii) size, (iii) concentration, and (iv) dispersion of the incorporated nanoparticles [18,19,20]. However, it is preferred to use homogeneously distributed nanoparticles [21]. It is also essential to consider a uniform distribution of nanoparticles because it profoundly affects the magnetic properties of nanofiber mats [22]. Furthermore, the effect of MNP (Fe_3_O_4_) concentration on drug loading, the encapsulation efficiency, and the release properties of the composite nanofibers play a pivotal role in the targeted delivery of various therapeutic agents [17]. On the other hand, electrospun Ni-based nanomaterials with tunable morphology and composition have been synthesized for various applications, such as electrochemical energy conversion, storage devices, and catalysis [23]. MNPs can respond significantly to the external magnetic field and have applications for cancer theragnostics [24,25]. However, there are shortcomings associated with the direct administration of MNPs intravenously or directly to the tumor, as they can leak from the target site due to their small size [16,26]. Polymeric nanofibers with embedded MNPs provide an effective platform for hyperthermia treatment. The nanofibers retain MNPs, thus, limiting their loss and ensuring better filling, leading to an enhanced magnetic response at the tumor site [27,28]. Moreover, polymer nanofibers can lead to prolonged drug delivery to cancer cells [29]. Thus, electrospinning can create functionalized MNFs with embedded MNPs to provide an ideal nanosystem for cancer treatment by ensuring localized delivery of iron oxide nanoparticles (IONPs) at the tumor site, thus, attracting considerable attention for the production of magnetic nanofibers.

Several investigations have combined the as-synthesized magnetic nanoparticles with various polymers to produce composite nanofibers through electrospinning [30,31]. In this study, we emphasized the fabrication of nanofibers embedded with magnetic particles of two distinct kinds, which provide different magnetic responses depending on the required applications. In particular, we are referring to biocompatible magnetic nanocomposites with zero residual magnetization if not activated, e.g., useful for magnetic scaffold [32] and magneto-thermal therapy [33], while on the other side, biocompatible systems with better magnetic and elastomagnetic performance to be employed in smart components [34]. Briefly, this investigation represents a fundamental study on the possibility of acquiring tunable magnetic properties by controlling the average particle size (d) and polydispersity index (PdI), as well as the degree of aggregation inside the polymer fibers. PdI is defined as the square of the standard deviation (*σ*) of the particle-size distribution divided by the average particle size: PdI = (*σ/d)^2^*. For this purpose, two types of magnetic systems were employed: (i) IONPs with an average size and polydispersity index of 7.2 nm and 3.3%, respectively, and (ii) nickel particles (NiPs) with a bimodal size distribution showing an average size of 129 nm and 600 nm at each mode, with a corresponding polydispersity index of 27.8% and 3.9%. Next, the synthesis of IONPs and the electrospinning methodology to produce functionalized MNFs are described thoroughly. The morphology and the magnetic response of the obtained fibers with embedded nanoparticles are also described. Furthermore, it is demonstrated that by controlling the particles’ morphology and adjusting the magnetic parameters accordingly, magnetic nanofiber systems exhibiting superparamagnetic behavior or ferromagnetic response can be produced. Specifically, the superparamagnetic status can be helpful when moderate magnetization needs to be applied and removed, i.e., for a magnetic scaffold [32]. On the contrary, the ferromagnetic one is essential to have a sufficiently high magnetization to obtain coupling with the mechanical properties of the fiber, i.e., due to the elastomagnetic effect [34,35].

## 2. Materials and Methods

### 2.1. Materials 

Oleic Acid (99%), 1-Octanol (>98%), and iron (0) pentacarbonyl (>99.99% trace metals basis) except hexadecylamine, which has a technical grade (90%), were purchased from Sigma Aldrich. All reagents were used without further purification. Commercial Nickel particles (purity 0.998, average nominal size less than 1 μm) were purchased from Sigma Aldrich, Saint Louis, MO, USA. Corethane, a medical-grade polycarbonate (Corvita Corporation, Miami, FL, USA), was used to prepare electrospun fibers. Organic solvents, including tetrahydrofuran (THF) and dimethylformamide (DMF), as well as other chemicals, such as Polyethylene glycol (PEG, Mw 700Da), were purchased from Sigma Aldrich, Saint Louis, MO, USA.

### 2.2. Synthesis and Characterization of IONPs

IONPs were synthesized through slight modifications in the process already reported in the literature [36]. Initially, 0.83 mmol of hexadecylamine and 6.33 mmol oleic acid were mixed and successively added to 8 mL of 1-octanol. The mixture was heated under continuous magnetic stirring up to 50 °C to homogenize the reagents. After cooling the solution to 23 °C, 13.81 mmol of Fe(CO)_5_ was added, and the mixture was then transferred in a 30 mL autoclave and subsequently heated to 200 °C with a slow ramp of 1.5 h, and the temperature was maintained for 5 h. After the reaction, the solution was washed with acetone through centrifugation at 6000 rpm for 10 min to remove the excess surfactant. For this purpose, 5 cycles of precipitation were performed, and the obtained product was dispersed in Toluene. Herein, we quantified the nanoparticle concentration through a well-known spectrophotometric method [37]. The quantification of the iron content is based on the coordination of three molecules of disodium 4,5-dihydroxy-1,3-benzenedisulfonate (tiron) at pH 7 with Fe^3+^ to form an extremely stable and strongly red-colored complex. The IONP suspension was digested with a mixture of nitric and hydrochloric acid. After complete digestion and oxidation of the iron ions to Fe^3+^, the residue was dissolved in HCl 0.1M, buffered with PBS, and added with Tiron. After the development of the complex [Fe(tiron)_3_]^3‒^, we recorded the UV-Vis spectrum in a 400–800 nm range. The detailed method and validation process are reported in a recent work [38]. The yield of synthesized MNPs was calculated to be 71%, which can be further optimized. Transmission electron microscopy (TEM) measurements were employed to determine the morphology and shape of the particles. TEM images were obtained using a Zeiss LIBRA 200FE-HRTEM operating at 200 kV equipped with a column Omega filter to increase the contrast. The sample was prepared by dropping 7 μL of solution on a copper grid and dried at room temperature (RT). Selective Area Electron Diffraction (SAED) pattern was obtained using the ITEM-TEM Imaging platform–Olympus Soft Imaging Solutions. More than 1500 IONPs were measured to obtain a distribution of equivalent diameters and their average value. This information was obtained using PEBBLES [Pebbles. Available online: http://pebbles.istm.cnr.it], software developed at the CNR [39].

### 2.3. NiP Morphological Analysis 

Transmission electron microscopy (TEM) was used to investigate NiPs’ morphology (i.e., size and shape of NiPs). TEM images were obtained using an FEI Tecnai G12 Spirit Twin, equipped with an LaB6 source and an FEI Eagle 4k CCD camera (Eindhoven, The Netherlands). The measurements were performed by applying an acceleration voltage of 120 kV. Prior to analysis, NiPs were dispersed in ethanol by sonication (DU-06, VEVOR, London, UK) for 5 min, then poured onto carbon-coated copper TEM grids.

### 2.4. Fabrication of Composite Electrospun Fibers 

Composite fibers were produced via electrospinning through a commercially available electrospinning setup (Nanon-01, MECC, Fukoaka, Japan). For fiber preparation, all chemicals and reagents were purchased from Sigma Aldrich. Briefly, Corethane, a medical-grade polycarbonate urethane (PCU, 15% *w*/*v*), was dissolved in a 50:50 solution of tetrahydrofuran (THF) and dimethylformamide (DMF) to form a homogeneous solution. Aliquot DMF was added to polyethylene glycol (10% *v*/*v*) to stabilize the dispersion of magnetic particles into the solution. Then, magnetic nanoparticles (NiPs and IONPs) at a concentration of 0.042 g/mL were efficiently mixed into the PCU solution for 20 minutes (approx.) until a homogeneous viscous solution with a uniform dark aspect was obtained. The solution was placed in a 5 mL plastic syringe connected to an 18-gauge needle. The fibers were randomly collected on grounded aluminum sheet using an optimized set of process parameters, e.g., 15 kV voltage; flow rate 1ml/h; needle/collector distance 150 mm. Fibers were collected for ca. 1 h by using a spinneret under translational motion (1 mm/s for a linear length of 120 mm).

### 2.5. Composite Electrospun Fiber Characterization

A preliminary assessment of the quality of the fiber processing was performed via field-emission scanning electron microscopy (FESEM) using FEI QUANTA200 (FEI, Hillsboro, OR, USA). In this case, samples were dried in a fume hood for 24 h to remove any residual solvent and sputter-coated with gold–palladium for about 20 s to obtain a 19 nm-thick conductive layer. SEM images were obtained under high vacuum conditions (10^−7^ torr) at 10 kV using the secondary electron detector (SED). Transmission Electron Microscopy (TEM) was performed to evaluate the spatial distribution of metal nanoparticles along the fibers. The fibers were collected by electrospinning for 60 s on a carbon-coated copper grid to obtain a few layers of fibers so that light could be transmitted easily. Bright-field TEM analyses were performed using an FEI TECNAI G12 Spirit-Twin microscope operating at 120 kV LaB6 source and equipped with an FEI Eagle 4k CCD camera. Particle-size distribution was performed on selected TEM images using image analysis freeware (NIH ImageJ 1.37). The magnetic properties of the composite fiber mats at RT were obtained from hysteresis loops recorded in a vibrating sample magnetometer (VSM, Matlab 9T, Oxford Instruments, Abingdon, UK) operating at a vibration frequency of 55 Hz and a fixed temperature of 300 K.

## 3. Results and Discussion

Figure 1a shows that the IONPs have a spherical shape, while Figure 1b illustrates the size distribution, with a mean diameter of 7.2 nm and a standard deviation of 1.3 nm, corresponding to a polydispersity index of 3.3%. It also specifies that the size distribution is uniform, symmetrical, and not too broad. SAED, reported in Figure 1c, is a typical powder pattern corresponding to the low-index reflection of spinel crystal structure, consistent with magnetite (Fe_3_O_4_) and maghemite (γ-Fe_2_O_3_). However, it is worth mentioning that it is difficult to distinguish these two structures by electron diffraction alone.

In Figure 2a, the TEM image shows a broad population of NiPs with a pseudo-spherical shape. In particular, Figure 2b shows a bimodal size distribution for NiPs with two average sizes of 129.0 ± 68.0 nm and 600.0 ± 119.0 nm, with a polydispersity index of 27.8% and 3.9%, respectively, and a relative volume fraction of 1:100, with the assumption of the spherical shape of NiPs. 

TEM images of the as-produced functionalized MNFs are shown in Figure 3a,c. In both cases, they refer to fibers that were produced by mixing a nominal volume fraction of MNPs equal to 15%. Noteworthily, the effective amount of NP into the fibers was significantly altered by the applied process conditions as a function of the characteristic size of NPs. As a preliminary step, loss of large particles was verified, as the solvent evaporation occurred during the electrospinning process, also confirmed by gravimetric analyses (data not shown), indicating an actual volume fraction of Ni NPs equal to 2%. Differences in size and volume fraction are clearly recognized in TEM images of both NP-loaded fibers.

In the case of IONP-loaded fibers (Figure 3a), it is evident that the incorporated MNPs form an agglomerate smaller than the fiber diameter and well confined within the fiber (as evidenced by the red boundary). In addition, this uniform distribution of MNPs within the agglomerates present in the fibers can also be related to the narrow size distribution of the MNPs themselves (good monodispersity), as shown in Figure 1b. 

Any agglomerate is well separated from the other, and it is impossible to distinguish every single MNP within it. However, in the case of NiP-loaded fibers (Figure 3c), the particles randomly aggregate within the fiber. Furthermore, as highlighted in Figure 3c, only the smallest particles (with an average size equal to 129.0 ± 68.0 nm) are trapped inside the polymer during the electrospinning process. The size distribution of NiPs remaining enclosed in the polymer, represented by the first mode of the distribution presented in Figure 2b, shows a larger polydispersity than IONPs. In comparison, larger particles (with an average size equal to 600.0 ± 119.0 nm) tend to preferentially escape from the fibers being formed as the solvent evaporates. Unlike the previous case, each particle is definitely distinguished from the other. Even though they are in contact, each grouping is quite distant from the other, with each cluster well covered by the polymer and enclosed in it. 

This corresponds to particle clusters embedded in the fiber, as shown in Figure 3c. Large agglomerates of different sizes and morphology are evident (green circles), while small particles can also be observed (yellow arrow), as suggested by the large size distribution (Figure 3c). In the case of particles smaller than 10 nm (as in the case of IONPs), it is evident that even though mechanical dispersion was not effective in separating them, aggregates of particles passed through the syringe’s needle and remained well embedded in the fibers. On the other hand, larger particles (as in the case of NiPs) tend to be partially removed from the fiber body (ca. 13%, data not shown), while, to a certain extent, given their ferromagnetic character, they tend to form large clusters.

Moreover, in many cases, these clusters tend to dilate the fiber itself, with smaller NiPs almost overflowing from it (red arrows). As shown in Figure 3b,d, the magnetic response of the IONP and/or NiP-loaded fiber mats is consistent with their average size and size distribution within the fibers. In particular, the behavior towards the saturation magnetization, with the same nominal volume fraction of the filling particles, occurs more rapidly for NiPs. Furthermore, in the case of NiP-loaded fibers, a coercive field of 0.2kOe, a remanence ratio (M_r_/M_s_) of 22%, and a constant magnetization above a magnetic field of 3kOe were measured. Conversely, in the case of fibers with embedded IONPs, we found an anhysteretic magnetization curve that continues to increase above a magnetic field of 3kOe. This is a typical shape of a RT hysteresis loop of similarly sized IONPs, i.e., typical superparamagnetic behavior. To understand the magnetic response of IONP-loaded fibers, it is essential to consider that the IONPs appear to be aggregated in TEM images of the fiber, but this does not mean that they are magnetically close to each other; in fact, the NPs are well separated from each other by a polymeric layer of which the fiber is made. Consequently, there is negligible interparticle exchange interaction between them. Moreover, considering the fact that the surface effects of single IONPs produce spin canting and a consequent decrease in their magnetic moment [40], a weak dipolar interaction is active between them. This indicates that the room-temperature magnetic response of IONP-loaded fibers tends to reflect the superparamagnetic behavior of IONPs, as expected for this size of IONPs. For fibers loaded with NiPs, the coercive field is similar to a single-domain or pseudo-single-domain state and more significant than a multi-domain state. Hence, the observed hysteresis loop correlates well with particle size approaching the critical size for single-domain or pseudo-single-domain behavior [41]. For a single-domain state, according to the Stoner and Wohlfarth model, M_r_/M_s_ = 0.5 results only if a random distribution of non-interacting uniaxial particles is present [42]. The particle magnetizations within the cluster are not parallel but point in slightly different directions. In the limit of strong exchange interactions, the cluster is uniformly magnetized, with a random easy axis. Therefore, each agglomerate behaves like a particle with uniaxial anisotropy and, since the uniaxial anisotropy of the agglomerates is random, the remanence ratio is given by the Stoner–Wohlfarth model (M_r_/M_s_ = 0.5). Probably, the difference between M_r_/M_s_ = 0.22 and the theoretical value predicted by the Stoner and Wohlfarth model is due to the presence of particles in the fibers that exhibit a mixture of both single-domain and multi-domain behavior (pseudo-single-domain), as suggested by the size distribution in the observed NiPs (see Figure 2b). Another probable cause of this reduction is the intercluster dipolar interaction [43]. Thus, it can be concluded that the NiP-loaded nanofiber mat produced by electrospinning approaches, fairly well, the ferromagnetic behavior due to single-domain or pseudo-single-domain NiP clusters interacting with each other. Therefore, tuning the chemical composition, the average size and polydispersity index of the magnetic particles embedded in the fibers allows for the production of fibers and mats with the desired morphology and magnetic properties, which can be useful in numerous applications. It is also worth noting that the mechanical properties of polymer composites depend on the polydispersity of the reinforcing fractions [44], and, in the case of electrospun membranes, the large aggregated particles act as local defect sites, thus, weakening the fibers [21].

## 4. Conclusions

This work presents a study concerning the magnetic response of nano/microparticle-loaded composite fiber mats, as a function of the peculiar properties, such as average size, polydispersity index, and aggregation state. For this purpose, two types of magnetic particles were used to produce composite MNFs: (i) IONPs with an average size of 7.2 nm and polydispersity index of 3.3% and (ii) NiPs with a bimodal size distribution, with an average size of 129 nm and 600 nm and polydispersity index of 27.8% and 3.9%, respectively. We verified that IONPs are well confined into the fiber body and tend to form regular agglomerates with characteristic sizes smaller than the fiber diameter. On the contrary, NiPs with average bigger sizes tend to accelerate the agglomeration phenomena, with the production of irregular clusters that induce a slight increase in fiber diameter, due to the effect of local deformation of the polymer matrix. We also demonstrate that the magnetic response of MNFs depends on fiber size and different characteristics of NPs (i.e., size, concentration, polydispersity index, and chemical composition). In particular, the collected results confirm the opportunity to fabricate magnetic nanofibrous systems with superparamagnetic and/or ferromagnetic behavior as a function of the average size, dispersion efficiency, and chemical composition of associated MNPs. In this perspective, the design of innovative membranes with different magnetic properties, as a function of microscopic (i.e., polydispersity index, chemical composition) and macroscopic (i.e., size, particle packing) properties of the magnetic filler could pave the way towards new insight for many applications in the biomedical field (i.e., multimodal theragnostic platforms, magneto-active systems for wound-healing stimulation).

## Figures and Tables

**Figure 1 jfb-14-00078-f001:**
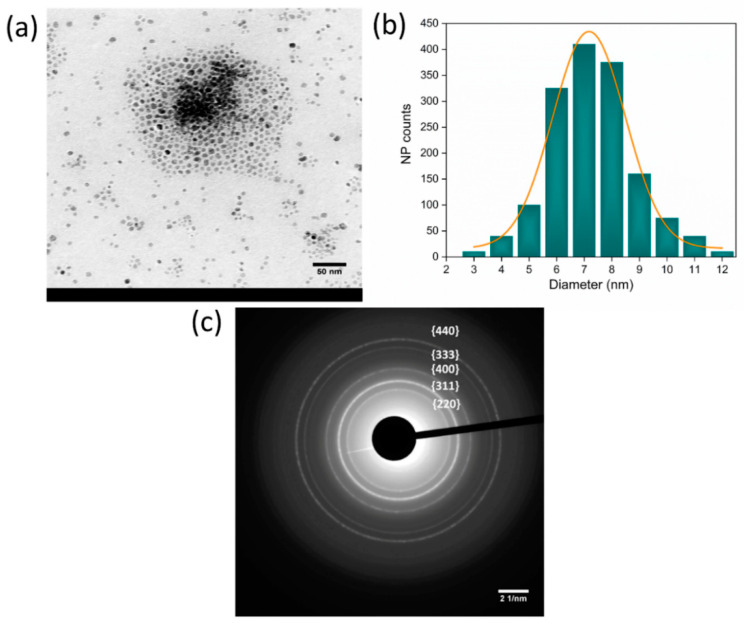
(**a**) TEM image of Fe_x_O_y_ nanoparticles; (**b**) the equivalent diameter distribution with the Gaussian fit indicates that the mean diameter of obtained MNPs is (7.2 ± 1.3) nm (the corresponding polydispersity index is 3.3%); (**c**) SAED pattern of the MNP sample showing different diffraction planes corresponding to the spinel crystal structure typical for maghemite and magnetite.

**Figure 2 jfb-14-00078-f002:**
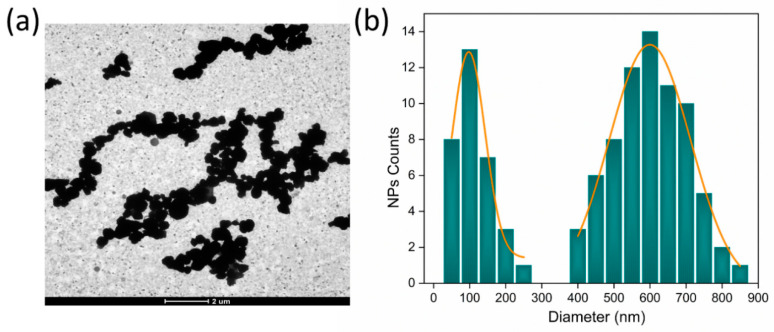
(**a**) TEM image and (**b**) NiP size distribution with the log-normal and Gaussian fit for the first and second mode of the distribution, respectively, indicating a mean diameter of 129.0 ± 68.0 nm and 600.0 ± 119.0 nm in each mode with a corresponding polydispersity index of 27.8% and 3.9%. The median value for the first mode of the size distribution is 114.0 nm.

**Figure 3 jfb-14-00078-f003:**
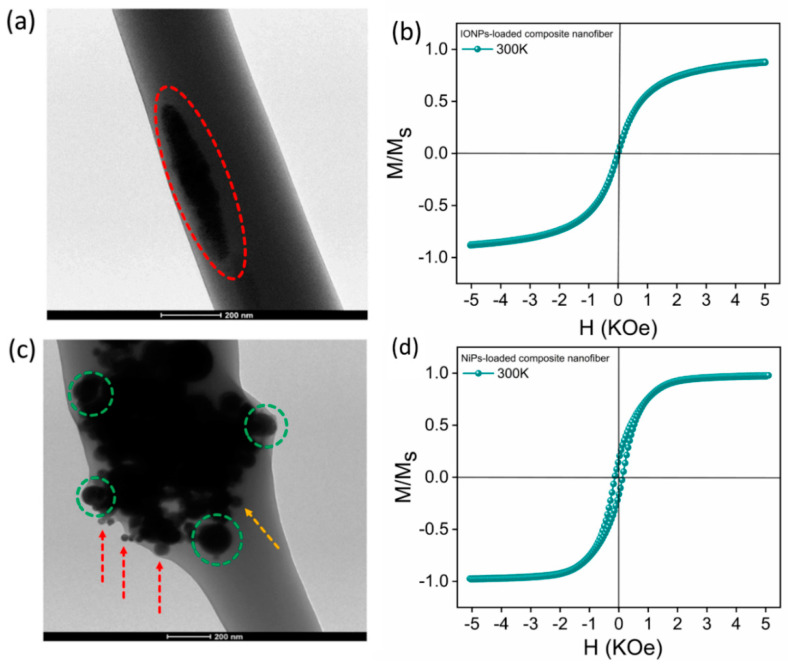
(**a**) Typical TEM image of an IONP-loaded composite nanofiber. The red circle defines a group of MNPs well embedded within the fiber; (**b**) hysteresis loop of an IONP-loaded nanofiber mat; (**c**) typical TEM image of an NiP-loaded composite nanofiber. The green circles indicate the presence of large agglomerates of different sizes/morphology, while yellow arrow indicates the presence of small particles, inside the fiber. Likewise, smaller NiPs that nearly overflow the fiber are specified by the red arrows; (**d**) hysteresis loop of NiP-loaded nanofiber mat. The nominal volume fraction of magnetic particles (IONPs or NIPs) is the same in both cases (15% by volume).

## Data Availability

Not applicable.

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
