# Peer review of "Magnetic Response of Nano/Microparticles into Elastomeric Electrospun Fibers"

_jfb, 2023, doi:10.3390/jfb14020078_

Round 1

Reviewer 1 Report

The article has many problems related to the English language, in my opinion. This is not a matter of being meticulous, but without this improvement, everyone will lose something. The Journal, the reader, and the authors since future citations will be precluded by the absence of readers. Thus, the authors should seek professional services for reading the work (which has good ideas and content) and correcting the format and general usage of the language.

Reviewer 2 Report

The manuscript submitted by V. Iannotti at la. reports on successful production, microscopic and magnetic characterization of electrospun  polycarbonate  urethane fibers with embedded magnetic nanoparticles. Two types of nanoparticles were used, namely (i) superparamagnetic iron oxide nanoparticles synthetized by the authors and (ii) commercial nickel nanoparticles revealing bimodal size distribution. In my opinion the results presented may be considered as a recipe for production of magnetic fibers by means of commercial ingredients. Thus, it may be of some interest for biomedical application.

Regarding the scientific quality of the manuscript I find it rather low due to lack of novelty as well as inconsistences and flaws listed below.

 1. Mean size of the smallest nanoparticles should rather be determined from log-normal distribution.

2. Determination of size for nanoparticles larger that 100nm using TEM is questionable due to lack of transparency. There are several other suitable methods for probing size of such particles, e.g. DLS or approach to magnetic saturation (provided that particles are single domain).

3. The method used for quantifying nanoparticle concentration (line 116 of the manuscript) should be explicitly described. Reference 37 is not useful.

4. The authors claim identical concentration of particles in both types of fibers, but do not present any proof. Moreover, considering that the concentration was identical in solution used for fiber production and that the large fraction of Ni nanoparticles is not present in the fibers, it is very likely that concentration is different in both case.

5. Magnetic characterization of pure particles should be presented along with that of the corresponding fibers. Magnetization values should be presented in proper units (not normalized) to allow assessment of particles' concentration.

6. Several statements is inconsistent thorough the manuscript:

- the mean size of the small fraction of Ni particles,

- "even distribution of MNPs" (line 190) vs. "agglomerates forming many particles" (line 201-212) vs. figure 3a vs. "aggregate particles are well separated from one another" (line 223),

- agglomerates in fibers with Ni nanoparticles to be discussed within S-W model (line 239), which is valid for non-interacting particles.

7. It is very unlikely that the statement of "constant magnetization above a magnetic field of 3kOe" (lines 219-220) is realized by physical system consisting of ferromagnetic (nonlinear M vs. H) and para/diamagnetic (linear M vs. H) fractions. To prove such conclusion the dependence of M vs. 1/H shall be plotted for the discussed field range.

8. The term "magnetic charges" (line 237) should be defined.

9. The conclusion of the last sentence of the abstract, namely "magnetic properties can be finely modulated by controlling the average size and polydispersity of magnetic particles" is not sufficiently supported by the results presented, especially regarding the adverb "finely". As such it should only be used in the discussion section (as possibility). It should also be discussed how to avoid possible problems in controling magnetic properties of fibers in view of incontrolled distribution of particles (especially the large ones).

Concluding, I could recommend this manuscript for publication only upon major revision that lifts all the inconsistences. It should include estimation of Ni particles' size using appropriate method, include magnetic characterization of the particles itself and discuss if these are maintained in the fibers (a prerequisite for controll of magnetic properties), and (strongly recommended) improves discussion of the results achieved in terms of possible applications.  

Round 2

Reviewer 2 Report

In the rebuttal letter I find responses to all the remarks of my review. I also appreciate the modifications to the manuscript  I find the resubmitted version as suitable for publication.